# Embedded GPU Implementation for High-Performance Ultrasound Imaging

Stefano Rossi  and Enrico Boni *

Department of Information Engineering, University of Florence, 50139 Firenze, Italy; stefano.rossi@unifi.it
* Correspondence: enrico.boni@unifi.it

**Abstract:** Methods of increasing complexity are currently being proposed for ultrasound (US) echographic signal processing. Graphics Processing Unit (GPU) resources allowing massive exploitation of parallel computing are ideal candidates for these tasks. Many high-performance US instruments, including open scanners like ULA-OP 256, have an architecture based only on Field-Programmable Gate Arrays (FPGAs) and/or Digital Signal Processors (DSPs). This paper proposes the implementation of the embedded NVIDIA Jetson Xavier AGX module on board ULA-OP 256. The system architecture was revised to allow the introduction of a new Peripheral Component Interconnect Express (PCIe) communication channel, while maintaining backward compatibility with all other embedded computing resources already on board. Moreover, the Input/Output (I/O) peripherals of the module make the ultrasound system independent, freeing the user from the need to use an external controlling PC.

**Keywords:** ultrasound; ULA-OP; high-speed communication; PCIe



## 1. Introduction

Ultrasound echography is becoming one of the most popular techniques for real-time medical imaging, not only for the absence of ionizing radiations but also for the continuous advancement of the related instrumentation [1,2]. A notable boost to such advancement has recently been given by the introduction of ultrasound open platforms [3,4], i.e., scanners that can be easily reconfigured to permit the implementation of imaging techniques based on novel transmission strategies and/or original processing algorithms [5–17]. Unfortunately, innovative algorithms frequently request a very high computational power to be implemented in real time, and this is not always possible with the current generation of research systems.

The so-called software-based open platforms [6,11,12] contain limited front-end electronics, and raw echo data are immediately digitized and streamed toward a PC where they are usually processed off-line by GPU boards such as, e.g., [18–20]. The main advantage of this approach is that the high computing power of GPUs is available to researchers having average knowledge of C language coding [21–24]. The limitation, however, is that data acquisition is usually performed in blocks and real-time operation is possible only for basic imaging modalities.

The ULtrasound Advanced Open Platform 256 (ULA-OP 256) is a compact open system developed for research purposes by the Microelectronics Systems Design Laboratory (MSDLab), of the University of Florence [25]. The scanner was designed to control up to 256 independent transmission and reception channels, allowing full control of linear (1-D) and small (2-D) probes. Due to the use of Field-Programmable Gate Arrays (FPGAs) and Digital Signal Processors (DSPs), ULA-OP 256 supports different real-time modalities and, at the same time, allows the acquisition of raw data as well of processed data at any point of the reception chain. ULA-OP 256 is managed by a modular and highly configurable software, running on the host PC, which initializes the hardware upon startup,

and, through a user-friendly interface, displays in real time the results of the ultrasound data processing. One disadvantage of this system is that the introduction of any new processing modality requires specific skills in the development of firmware based on different programming languages.

This work aims to increase the processing capability of ULA-OP 256 by including onboard GPU resources. A new expansion board (EXP) hosting a Jetson AGX Xavier module (NVIDIA, Santa Clara, CA, USA) [26] was designed and implemented; see Figure 1. By exploiting the embedded GPU resources, this upgrade will allow the performance of the most recent processing modalities that require a massive amount of computational resources [27,28].

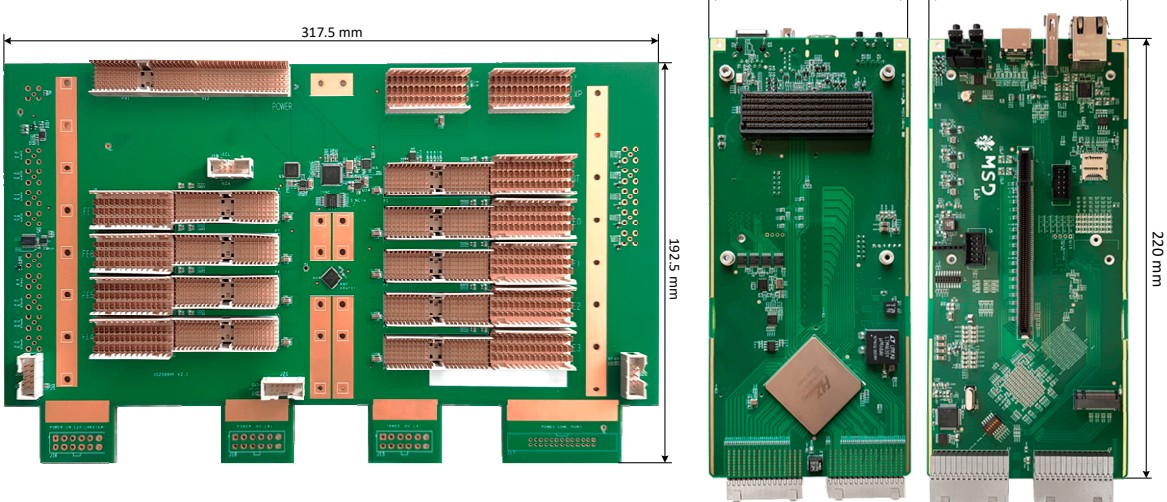

**Figure 1.** On the left, top view of the Back-Plane (BKP) board; on the right, top and bottom view of the expansion (EXP) board without the Jetson module assembled on it.

## 2. System Design

The current architecture of the ultrasound system is composed of 8 Front End boards (FEs), each managing 32 transmit/receive (TX/RX) channels, and one Master Control (MC) board, which are interconnected through the Back-Plane (BKP) board. Each FE is also connected to 32 probe elements [29] and integrates an FPGA from the ARRIA V GX Family (Altera, San Jose, CA, USA), which is programmed to: (1) generate high-speed bit streams that correspond to arbitrary TX signals [30]; (2) amplify and sample (at 78.125 MHz) the RX echo signals; and (3) perform the first stage of the delay and sum beamforming [31]. Furthermore, the same boards host two 8-core DSPs (TMS320C6678, Texas Instruments, Austin, TX, USA) running at 1.2 GHz, and 8 GB of DDR3 memory. These DSPs are in charge of real-time processing operations, such as coherent demodulation, filtering, and compounding.

The MC board coordinates the FE boards through a SerialRapidIO (SRIO) ring bus, which can reach a full-duplex transfer rate of 10 GB/s for each FE, and manages the communication to the host PC though Ethernet (ETH) or USB 3.0 SuperSpeed.

The "add-on" EXP board was connected with the MC DSP and the FE DSPs through a new PCIe star network. The PCIe protocol was characterized by a full-duplex point-to-point topology. The peripheral devices referred to as End-Points (EPs), are connected to the host, named Root Complex, through switches which perform the function of nodes. The connections are made up of pairs of differential TX and RX lanes. The number of lanes per link can range from 1 to 32 ($\times$1, $\times$2, $\times$4, $\times$8, $\times$12, $\times$16, and $\times$32) and is automatically negotiated during the initialization process. Moreover, the protocol does not provide limitations on the simultaneous access between multiple EPs. The link performance of

some PCIe configurations is reported in Table 1. The higher throughput of the PCIe3.0 protocol is mainly due to the different types of encoding.

**Table 1.** PCIe throughput specifications.

| PCIe Version | Encoding | Transfer Rate (GT/s) | Throughput (GB/s) | | | | |
|:---:|:---:|:---:|:---:|:---:|:---:|:---:|:---:|
| | | | ×1 | ×2 | ×4 | ×8 | ×16 |
| 2.0 | 8b/10b | 5.0 GT/s | 0.5 | 1 | 2 | 4 | 8 |
| 3.0 | 128b/130b | 8.0 GT/s | 0.985 | 1.969 | 3.938 | 7.877 | 15.754 |

The project required the update of the BKP design, on the left side of Figure 1, to host 204 differential lanes with controlled impedance over 14 layers: 66 PCIe connections towards the PCIe switch located on the EXP, in red in Figure 2; together with the 138 lanes of the SRIO ring bus, in light blue in Figure 2. These changes were designed to ensure backward compatibility with the previous system configuration, thus keeping the SRIO ring bus available.

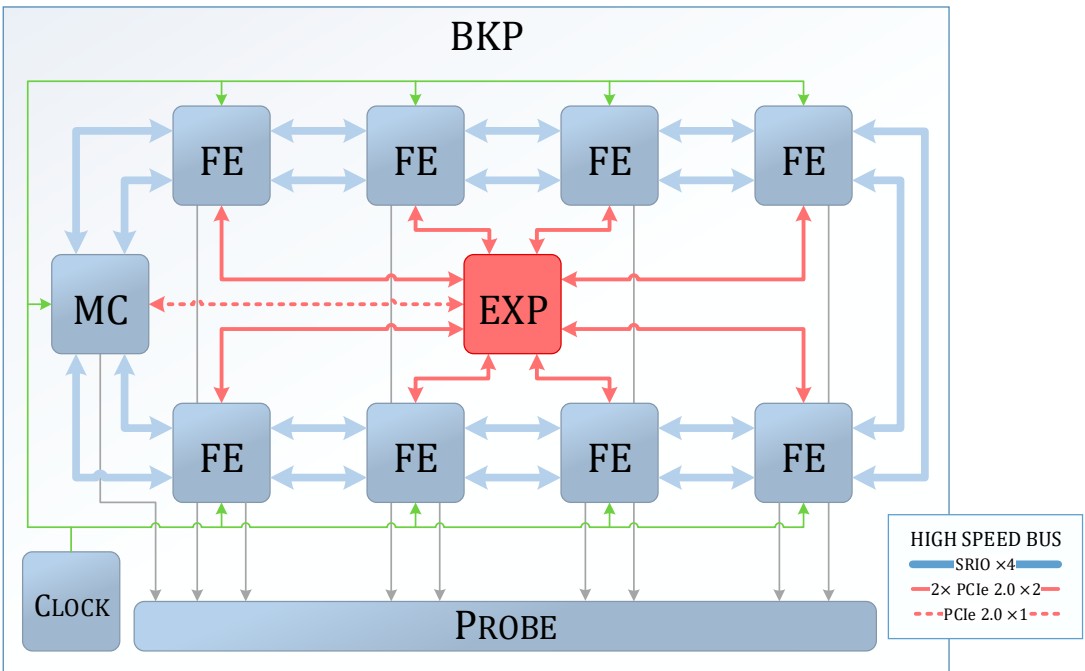

**Figure 2.** The modified Back-Plane architecture of ULA-OP 256. The new EXP board and related connections are highlighted in red. The Master Control (MC) and the Front End (FE) boards are also connected to the ultrasound probe.

### 2.1. Expansion Board

The main resources of the EXP board are shown in Figure 3. The core of the board is represented by the PCIe switch PEX8796 (Broadcom Limited, Irvine, CA, USA), capable of managing up to 96 PCIe3.0 lines, distributed in 6 stations of 4 ports each (natively ×4). The ports of the same station can be combined to obtain configurations up to ×16. The device can interconnect ports configured with different numbers of lines and guarantees backward compatibility with PCIe 2.0 standard.

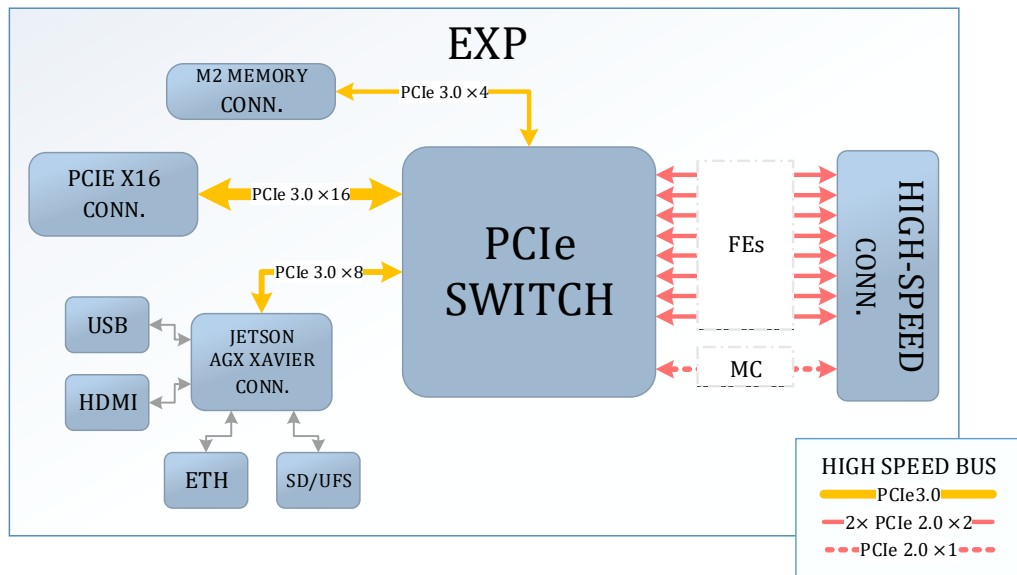

**Figure 3.** Peripheral Component Interconnect Express (PCIe) connection diagram of the expansion board. The PCIe2.0 and PCIE3.0 buses are shown in red and yellow lines, respectively.

The diagram of the PCIe connections is highlighted in Figure 3:

- 17 ports are reserved for the DSPs integrated on the FE and MC modules (1 port for each DSP).
- An M2 slot PCIe3.0 ×4 (1 port) can host a common consumer SSD (solid-state drive) NVMe (non-volatile storage memory express) storage with a capacity of up to 2 TB.
- A PCIe3.0 ×16 connector (4 ports) connects to additional PCIe external resources.
- The Jetson AGX Xavier module is connected through a PCIe3.0 ×8 bus (2 ports)

The latter is a system-on-module (SOM) which integrates 512 CUDA cores Volta GPU (64 tensor cores) at 1.37 GHz per core. An embedded Linux operating system runs on an Octal-core NVIDIA Carmel ARMv8.2 CPU with 32 GB LPDDR4x of RAM and 32 GB of eMMC 5.1 storage. Furthermore, the EXP implements onboard peripherals to independently control the Jetson module, specifically an Ethernet connector (gigabit compliant), USB 3.0, HDMI 2.0 (as shown in Figure 4.), and an SD/UFS socket.

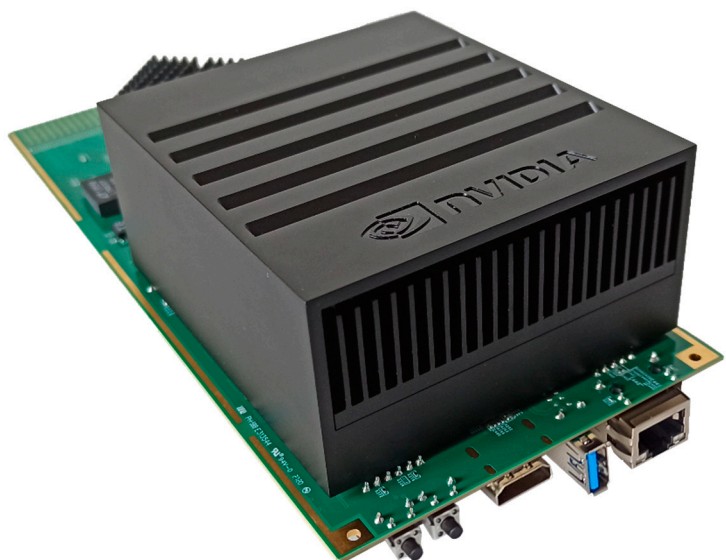

**Figure 4.** Isometric view of the expansion board. In the foreground, the Jetson module installed on the custom board, with HDMI, USB and gigabit Ethernet connectors.

### 2.2. Root Complex Configurations

The architecture developed in this system allows setting the Root Complex to be either the Jetson or the MC DSP, Figure 5. This second configuration is meant for those cases in which the use of the GPU module is not required, and the role of the PCI Express tree is to expand the interconnection between the FEs and the MC. However, according to the PCIe protocol, as soon as the network is instantiated, any device can initiate a transaction with any other device. It should be noted that all resources do not always need to be connected. The PCIe switch automatically detects which are disconnected by isolating the corresponding ports from the network.

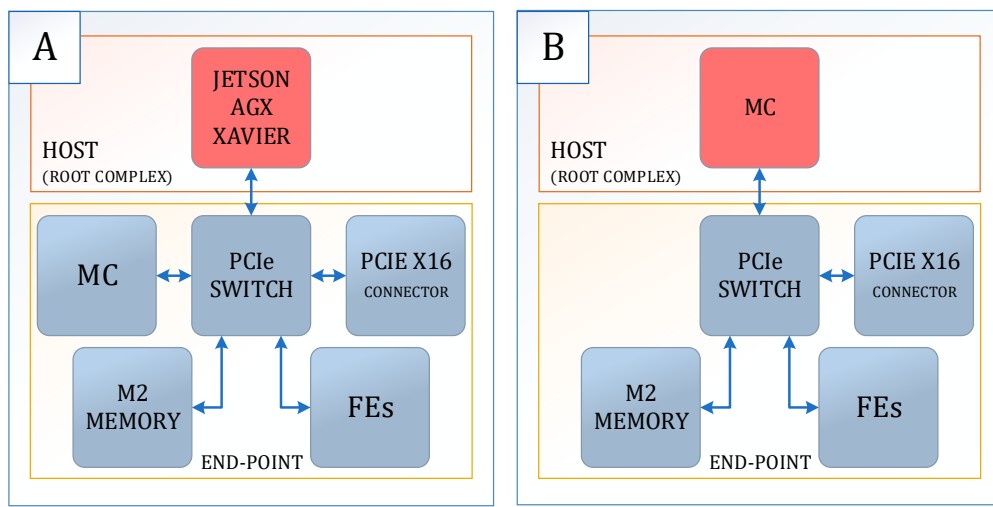

**Figure 5.** The single host architecture with 2 different Root Complex (Host) scenarios: (**A**) with Jetson AGX Xavier and (**B**) with Master Control (MC) board.

## 3. Results and Discussion

### 3.1. PCIe Bandwidths and Xavier AGX Performances

Table 2 summarizes the theoretical throughputs of the PCIe resource channels onboard ULA-OP 256 by the new design. As highlighted in Figure 3, both DSPs of each FE board are connected to the PCIe switch with a PCIe2.0 $\times 2$ connection. Therefore, the total 16 DSPs, that work in parallel on the acquired data, reach a throughput of about 16 GB/s, while the bandwidth of the single DSP of the MC card, which is a PCIe 2.0 $\times 1$ link, is 0.5 GB/s. The bandwidth of the PCIe x16 connector is comparable to that of the FEs (around 16 GB/s) and is twice that available for the Jetson module. For the M2 memory, a throughput of about 4 GB/s was considered when the SSD supports PCIe 3.0 $\times 4$. To evaluate the effective bandwidth available in TX and RX on the DSPs, preliminary tests were conducted by placing the devices in direct communication. The results obtained are consistent with those carried out in previous works on the TMSC6678 device [32]. Protocol and implementation overhead result in an additional loss of bandwidth; the throughput was thus estimated at 0.75 GB/s for each FE DSP. However, the overall bandwidth of 12 GB/s is much greater than that available for Jetson, especially assuming a realistic reduction in the actual PCIe bandwidth for the Nvidia module of about 30% (about 6 Gbit/s). As for the processing performance achievable through the Jetson processing module, the 512 CUDA cores can reach a theoretical limit of 1.4 TeraFLOPS (FLoating Point Operation Per Second) with single-precision floating point numbers (FP32) which correspond to 0.7 TeraFLOPS with double-precision numbers (FP64). In addition, the Volta architecture allows us to achieve up to 22.6 TeraOPS (Tera Operations Per Second) with 8-bit fixed point precision, or 11.3 TeraFLOPS with half-precision floating point (FP16), due to the 64 tightly coupled tensor cores, additional matrix multiplication, and accumulation operations that can run parallel with the other CUDA cores [33]. To compare the performance of AGX Xavier with previous Jetson board models, we can refer to the benchmarks on imaging applications

that require low latency [34]. For example, when comparing the performance of a generic jpg encoding test (1920 × 1980, 90%, 4:4:4), the execution time obtained by the Xavier AGX (0.75 ms) was 4.9 and 4.1 times shorter than previous generation TX1 and TX2 modules, respectively.

**Table 2.** Theoretical Resources Throughput.

| Resources | Protocol | Number of Lanes | Throughput (GB/s) |
|-----------|----------|-----------------|-------------------|
| FEs DSPs | PCIe2.0 | 16(DSPs) ×2 | 16 |
| MC DSP | PCIe2.0 | ×1 | 0.5 |
| M2 | PCIe3.0 | ×4 | 3.938 |
| PCIe 16x Conn. | PCIe3.0 | ×16 | 15.753 |
| Jetson Xavier | PCIe3.0 | ×8 | 7.876 |

### 3.2. Application Examples

Here, we will consider two different application examples. In the first, it was assumed that no preprocessing operations (beamforming, demodulation, filtering, etc.) were performed by the FPGAs and DSPs available on FE boards during the RX phase. In the second application, however, the data were preprocessed by the FE boards using the embedded FPGA beamformer and the demodulation and filtering capabilities of the DSPs, while the GPU cores were exploited to evaluate the performance of a vector velocity estimation method.

Let us consider the Jetson as Root Complex (RC) (Figure 5A). In the first example, the raw data were directly transferred to the Jetson module through the PCIe channel. Figure 6 shows the color-coded image of the band saturation percentage, which correlates with the number of active receiving channels, the pulse repetition frequency (PRF), the raw data decimation (RDD), and the acquisition time (AT), i.e., the sampling interval within two pulse repetition interval (PRI) events. Supposing data were acquired from 256 channels with a pulse repetition frequency (PRF) of 5 kHz, a depth range of 30 mm (i.e., an acquisition time of 40 µs), and an RDD of 1, the band saturation would stand at 90%. Nevertheless, a PRF of 5 kHz could, in theory, allow a non-ambiguous range of 14.6 cm, requiring an acquisition time equal to 200 µs. In this case, to acquire the whole range of depths, either the sampling frequency or the number of elements should be reduced. It is worth noting that the image shown in Figure 6 highlights that dividing the sampling frequency by 5 (RDD = 5), down to 15.6 MHz, would be sufficient to theoretically obtain all the available depths without any limit on PRF and number of active channels. The CUDA core performance of 1.4 TeraFLOPS allows the Xavier module to perform real-time beamforming. Assuming we beamform the incoming data on the Jetson, we consider the delay and sum algorithm [35], which performs a total number of sums and multiplications per frame approximately equal to the number of receiving elements multiplied by the sample gates and by the number of beamformed lines computed in a PRI. Any single-precision addition and multiplication can be considered a FLOP. Considering the computing performance, the system can beamform all 256 elements of ULA-OP, with 1000 gates per line, producing 10,000 frames per second, each frame being composed of 256 scan lines.

In the second example, it was assumed to shift the beamforming and demodulation processes to the FE FPGAs and DSPs of ULA-OP256, from the GPU cores, and to perform additional algorithms, such as a vector Doppler method [27]. A preliminary implementation of this algorithm was developed to efficiently exploit Jetson Xavier's GPU resources. Each frame was subdivided into partially overlapped estimation blocks of samples (kernels) in order to obtain an output estimates array for both blood velocity components (x and z directions). The method was tested on 50,000 RF frames, bufferized, and cyclically transferred and processed on GPU. The size of output estimates array, and the dimension of the estimation kernel were varied by evaluating the processing time. Table 3 shows that

the available real-time frame rate is 2.5 kHz for the slower case, with maximum kernel size and number of estimator outputs, and increases of about 40%, reaching 3500 kHz, decreasing the estimates and the size of the kernels. The frame rate does not scale linearly with the output size, because part of the processing time is due to the preliminary steps of the processing chain, which provide for the filtering of all input data, regardless of the number of kernels, kernel size, and estimation arrows chosen. It should be noted that having a real-time estimation with such a high frame rate, which for human vision would be near 50/60 Hz, is still useful for applying additional processing modes to improve image quality.

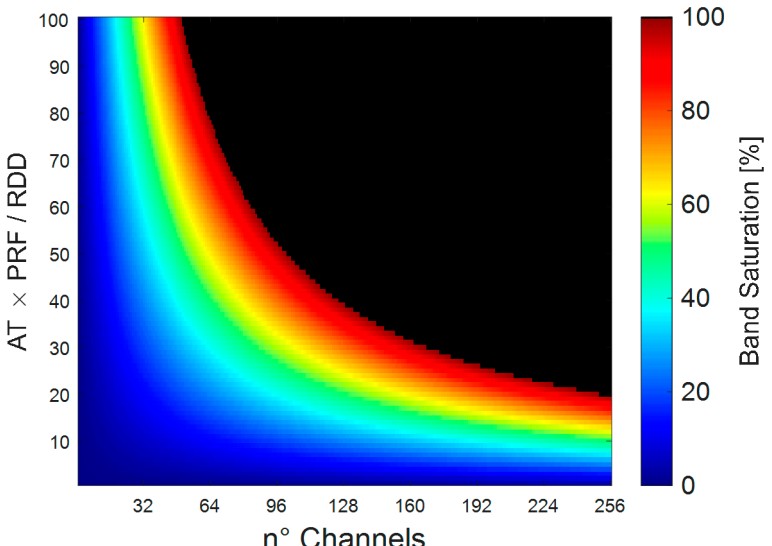

**Figure 6.** Expected saturation level of PCIe 3.0 ×8 connection towards NVIDIA Jetson AGX Xavier. The transfer band saturation percentage was color coded.

**Table 3.** Vector Flow Imaging Method Processing Time Test.

| Kernel Size (Samples) | | Output Estimates Matrix (Estimations Per Frame) | | Real-Time Frame Rate (Hz) |
|---|---|---|---|---|
| z-dim | x-dim | z-dim | x-dim | |
| 50 | 10 | 50 | 20 | 3512 |
| 100 | 10 | 50 | 20 | 3423 |
| 50 | 20 | 50 | 20 | 3339 |
| 100 | 20 | 50 | 20 | 3396 |
| 50 | 10 | 100 | 20 | 3203 |
| 100 | 10 | 100 | 20 | 3034 |
| 50 | 20 | 100 | 20 | 3104 |
| 100 | 20 | 100 | 20 | 2956 |
| 50 | 10 | 50 | 40 | 3200 |
| 100 | 10 | 50 | 40 | 3094 |
| 50 | 20 | 50 | 40 | 3092 |
| 100 | 20 | 50 | 40 | 3026 |
| 50 | 10 | 100 | 40 | 2717 |
| 100 | 10 | 100 | 40 | 2641 |
| 50 | 20 | 100 | 40 | 2586 |
| 100 | 20 | 100 | 40 | 2488 |

## 4. Conclusions

A new system architecture, including PCI-Express interconnection, has been implemented. The Jetson Xavier processing module adds processing capabilities to the previous

system and will allow the implementation of a fully embedded system that does not need an external controlling PC, due to the integrated display peripherals and USB connections for keyboard and mouse.

Although the bandwidth available from the FE DSPs to the Jetson Xavier is just under one-half that of the sum of the peak data rate of all the FE boards, it is still very large with respect to the current interconnection interface of the UO256 system, USB 3.0. Both the raw data throughput limits, i.e., without the onboard FE processing, and an example of vector Doppler imaging application, exploiting the beamforming, demodulation, and decimation performed by the FE resources of ULA-OP 256, have been addressed. The results of this study show that the system in this configuration is able to perform the aforementioned tasks. Finally, the additional PCIe3.0 x16 expansion slot allows the system to integrate additional GPU resources, further enhancing processing performance. This new architecture will, therefore, allow the implementation of a new class of computationally demanding algorithms directly inside the system, overcoming the burden of external data transfer.

**Author Contributions:** Conceptualization, S.R., and E.B.; Formal Analysis S.R. and E.B.; Investigation S.R. and E.B.; Methodology, S.R.; Visualization, S.R.; Writing—Original Draft, S.R.; Writing—Review and Editing, E.B.; Supervision, E.B. Both authors have read and agreed to the published version of the manuscript.

**Funding:** This work is part of the Moore4Medical project funded by the ECSEL Joint Undertaking under grant number H2020-ECSEL-2019-IA-876190.

**Acknowledgments:** The authors are grateful to Piero Tortoli for his helpful suggestions and guidelines.

**Conflicts of Interest:** The authors declare no conflict of interest.

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
