# Peer review of "Embedded GPU Implementation for High-Performance Ultrasound Imaging"

_electronics, doi:10.3390/electronics10080884_

Round 1

Reviewer 1 Report

This is a highly technical paper describing the integration of NVidia Jetson kit into the UPLA-OP system.  From my perspective, it's more a whitepaper focusing on the PCI-E interconnection of all modules with the Nvidia kit than a scientific paper. I somehow miss information about the software part of the project (firmware, communication protocols, etc.) and better evaluation of the performance. I did also not learn whether the system is already alive or the presented results are only theoretical estimations.

Reviewer 2 Report

In the introduction, authors note that „A notable boost to such advancement has been recently given by the introduction of ultrasound open platforms [3]“. This statement is based only one reference to their own publication. In the next paragraph analogous situation „The so-called software-based open platforms [3] contain limited front-end electronics, and raw echo-data are immediately digitized“. There are other Ops in the field: SARUS, SonixTouch, Verasonics and etc.

[3] Boni, E.; Yu, A.C.H.; Freear, S.; Jensen, J.A.; Tortoli, P. Ultrasound Open Platforms for Next-Generation Imaging Technique Development. IEEE Transactions on Ultrasonics, Ferroelectrics, and Frequency Control 2018, 65, 1078–1092, doi:10.1109/TUFFC.2018.2844560

30% of references are authors’ self-citation [3,9, 10, 17, 18, 20,21, 23, 28, 29]. It is inappropriate and violates the rules of MDPI „Authors should not engage in excessive self-citation of their own work“.

I don’t think that table 2 with the technical specification of Nvidia Jetson AGX Xavier (available on the internet, i.e., https://www.nvidia.com/en-us/autonomous-machines/embedded-systems/jetson-agx-xavier/ ) should be included in the manuscript. By the way, majority of parameters are repeated in the text (lines 92-97).

More experimental results should be carried out in order to validate the new architecture.

Reviewer 3 Report

A new system architecture, including PCI-Express interconnection, has been implemented. The Jetson Xavier processing module adds processing capabilities to the previous system and will allow the implementation of a fully embedded system that doesn’t need an external controlling PC, thanks to the integrated display pheripherals and USB connections for keyboard and mouse. However, I think the case study is not sufficient to verify the proposed system. More test and evaluation must be performed to illustrate the feasibility and possibility of the new system.

Round 2

Reviewer 2 Report

Updated manuscript version looks much better better than the previous version.

Reviewer 3 Report

Good work